# On Using CFD and Experimental Data to Train an Artificial Neural Network to Reconstruct ECVT Images: Application for Fluidized Bed Reactors

Carlos Montilla [1,†], Renaud Ansart [1,*,†], Anass Majji [2,†], Ranem Nadir [2,†], Emmanuel Cid [1,†] , David Simoncini [2,†] and Stephane Negny [1,†]

1  Laboratoire de Génie Chmique, Université de Toulouse, CNRS, INPT, UPS, 31030 Toulouse, France; emmanuel.cid@ensiacet.fr (E.C.)
2  IRIT UMR 5505-CNRS, Université de Toulouse, 31030 Toulouse, France; anass.majji@etu.toulouse-inp.fr (A.M.)
*  Correspondence: renaud.ansart@toulouse-inp.fr
†  These authors contributed equally to this work.

**Abstract:** Electrical capacitance volume tomography (ECVT) is an experimental technique capable of reconstructing 3D solid volume fraction distribution inside a sensing region. This technique has been used in fluidized beds as it allows for accessing data that are very difficult to obtain using other experimental devices. Recently, artificial neural networks have been proposed as a new type of reconstruction algorithm for ECVT devices. One of the main drawbacks of neural networks is that they need a database containing previously reconstructed images to learn from. Previous works have used databases with very simple or limited configurations that might not be well adapted to the complex dynamics of fluidized bed configurations. In this work, we study two different approaches: a supervised learning approach that uses simulated data as a training database and a reinforcement learning approach that relies only on experimental data. Our results show that both techniques can perform as well as the classical algorithms. However, once the neural networks are trained, the reconstruction process is much faster than the classical algorithms.

**Keywords:** ECVT; 3D ECT; fluidization; deep learning; multi-phase flow

## 1. Introduction

Nowadays, fluidized beds play a major role in many industrial processes. They offer numerous advantages that are great for gas-solid mixing and uniform conditions inside the reactor. These characteristics are, however, extremely dependent on the internal behavior and dynamics of the fluidized bed. Due to their nature, it is very difficult to obtain experimental information inside the reactor. Probes and pressure taps provide only local information and could be intrusive and disturb the internal dynamics. Optic techniques are more difficult due to the opaque properties of most particles, and they are very difficult to implement in a 3D reactor. Electrical capacitance volume tomography (ECVT or, more precisely, 3D ECT) is a measurement technique that addresses all these problems [1]. This device is capable of reconstructing the internal 3D solid volume fraction distribution using only capacitance sensors located externally at the walls of the reactor. Therefore, it is a fully non-intrusive and non-invasive technique that can provide valuable information to characterize the reactor and even validate the CFD simulations and mathematical models [2–6]. The key parameter of the ECVT system is the reconstruction algorithm used to obtain the 3D solid volume fraction distribution from the capacitance measurements. Several propositions can be found in the literature. Recently, an approach using machine learning and artificial neural networks has been studied, and it has proved to be very performant [7]. Because they do not rely on an iterative algorithm to reconstruct an image, they are much faster than the classical reconstruction approaches. This makes them suitable when we

need to process data in real time or when the amount of images to process is very large. However, the accuracy of these approaches is directly linked to the training database used. Previous works have used very limited or simple datasets to train their neural network as they were more interested in the reconstruction algorithm itself and not in their application to fluidized beds [8]. In this work, we propose a strategy to generate a training database that could be adapted to any kind of configuration. We will study two different ways of generating suitable data to train an artificial neural network for the image reconstruction problem in ECVT systems. The first approach consists of extracting the information from accurate 3D numerical simulations to simulate a fluidized bed with an ECVT system, assuming the sensitivity matrix approximation. This will enable us to obtain a great amount of data on fluidized-bed-like patterns that can be used to train our neural network. The second approach directly uses the experimental data coming from the experimental device by using sensitivity matrix approximation, and no previous knowledge of the volume fraction distribution is required. This second approach is completely self-sufficient and can take advantage of any new experimental data to improve its quality further.

## 2. Electrical Capacitance Volume Tomography (ECVT)

Electrical capacitance volume tomography is a non-invasive and non-intrusive device capable of mapping the 3D solid volume fraction distribution inside a sensing region. It consists of a series of electrodes placed around the interest volume. By measuring the $m$ inter-sensor capacitances between all pairs of electrodes, we can obtain an approximation of the solid volume fraction ($\boldsymbol{\alpha}_p$) in a series of $n$ predefined voxels inside. Using sensitivity matrix approximation, we can establish the following relationship between the capacitance vector and the solid volume fraction vector, known as the forward problem:

$$\mathbf{C} = S\boldsymbol{\alpha}_p \tag{1}$$

where $\mathbf{C} \in \mathbb{R}^{m \times 1}$ is a vector with the capacitances measurements, $\boldsymbol{\alpha}_p \in \mathbb{R}^{n \times 1}$ is a vector containing the solid volume fraction values at the $n$ different voxels, and $S \in \mathbb{R}^{m \times n}$ is called the sensitivity matrix.

Numerous approaches can be found in the literature that try to approximate a solution for Equation (1). The first approaches studied consisted mainly of optimization problems that needed to be solved using iterative methods [9]. Although these methods could be very accurate, the iterative process can be computationally expensive, especially if we need to analyze a large number of images or if we need to obtain results in real time. For this reason, a new type of algorithm has been developed using machine learning techniques [7,8]. These approaches train artificial neural networks so they can learn how to reconstruct images coming from an ECVT system.

In this work, we used an ECVT system developed by Tech4Imaging. The system is composed of 36 electrodes distributed radially in four rows of nine electrodes. Each electrode is 19 mm tall and has an arc of 32° (see Figure 1). The sensing volume corresponds to a cylinder 10 cm in diameter and 10 cm in height with an acquisition frequency of 50 Hz. This sensing region is divided into 8000 voxels. The mean signal-to-noise ratio (SNR) of the 630 channels is 39.9775 dB, and 530 channels have an SNR above 30 dB.

The sensitivity matrix was provided by Tech4Imagning, and it was calculated using COMSOL simulations. The simulations are built by setting one of the voxels to a reference fill state and all other voxels to the void state; then, the inter-plate capacitance is measured by exciting each of the electrodes one by one. This process is repeated for each voxel in the domain. This methodology neglects the impact of the gas flow in the sensitivity matrix, which is a source of error in the model.

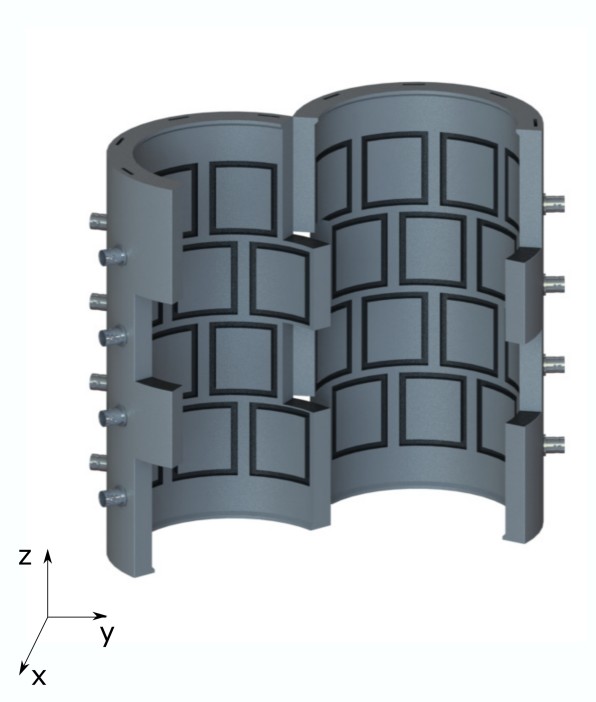

**Figure 1.** ECVT system, consisting of 36 electrodes 19 mm in height with 32° arcs, organized in four rows of nine electrodes each.

In order to test the accuracy of the reconstruction algorithms found in the literature, we used the ECVT device in a fixed bed of glass beads with an empty spherical object inside. Several object diameters were studied, with the smallest object being 2.7 cm in diameter and the biggest object being 5.5 cm in diameter. These objects were chosen as they are representative of the bubbles we would obtain in our fluidized bed, which would vary between 2 cm–6 cm and would rise at a frequency of 1.5 Hz–2.5 Hz. Figures 2 and 3 show the reconstructed images for the two extreme cases for three different algorithms: the linear back projection algorithm (LBP), the iterative linear back projection algorithm (ILBP), and the multi-objective image reconstruction technique (MOIRT) [10,11]. We can see that the LBP algorithm fails to reconstruct the void object inside the domain completely, and even the smallest object is not seen at all. The other two algorithms successfully reconstruct both void objects with a spherical-like shape. In order to check that the reconstructed objects have the correct size, we calculated the equivalent diameters based on the area of the iso-surface at $\alpha_p = 0.3$ (Equation (2)). Table 1 shows the reconstructed diameters. We notice that the ILBP and MOIRT algorithms can accurately predict the size of the biggest objects, but they underestimate the size of the smallest objects, especially the ILBP. Moreover, the reconstruction processes were performed in a computer with 32 cores using the Intel CPU Intel(R) Xeon(R) CPU E5-2620 v4 @ 2.10GHz with 32 GB RAM memory. When using these resources, the reconstruction time for one single image was 50 ms for the LBP algorithm, 2.2 s for the ILBP algorithm, and 25 min for the MOIRT. The ILBP and MOIRT reconstruction times are too high for applications where real-time monitoring is needed or when analyzing a large amount of data. For reference, reconstructing 5 min worth of data with our ECVT device would take more than 9 h with the ILBP algorithm and more than 8 month with the MOIRT algorithm, but it would take only 12 min with the LBP algorithm. It is for this reason that we chose to explore the use of a neural network-based approach to obtain a fast and accurate reconstruction algorithm.

$$d_{object} = \sqrt{\frac{A_{object}}{\pi}} \tag{2}$$

**Table 1.** Equivalent diameter of the reconstructed objects using the classical algorithms.

| $d_{obj}$ | 55 mm | 50 mm | 44 mm | 40 mm | 27 mm |
|---|---|---|---|---|---|
| LBP | × | × | × | × | × |
| ILBP | 57 mm | 53 mm | 49 mm | 42 mm | 18 mm |
| MOIRT | 56 mm | 54 mm | 49 mm | 43 mm | 23 mm |

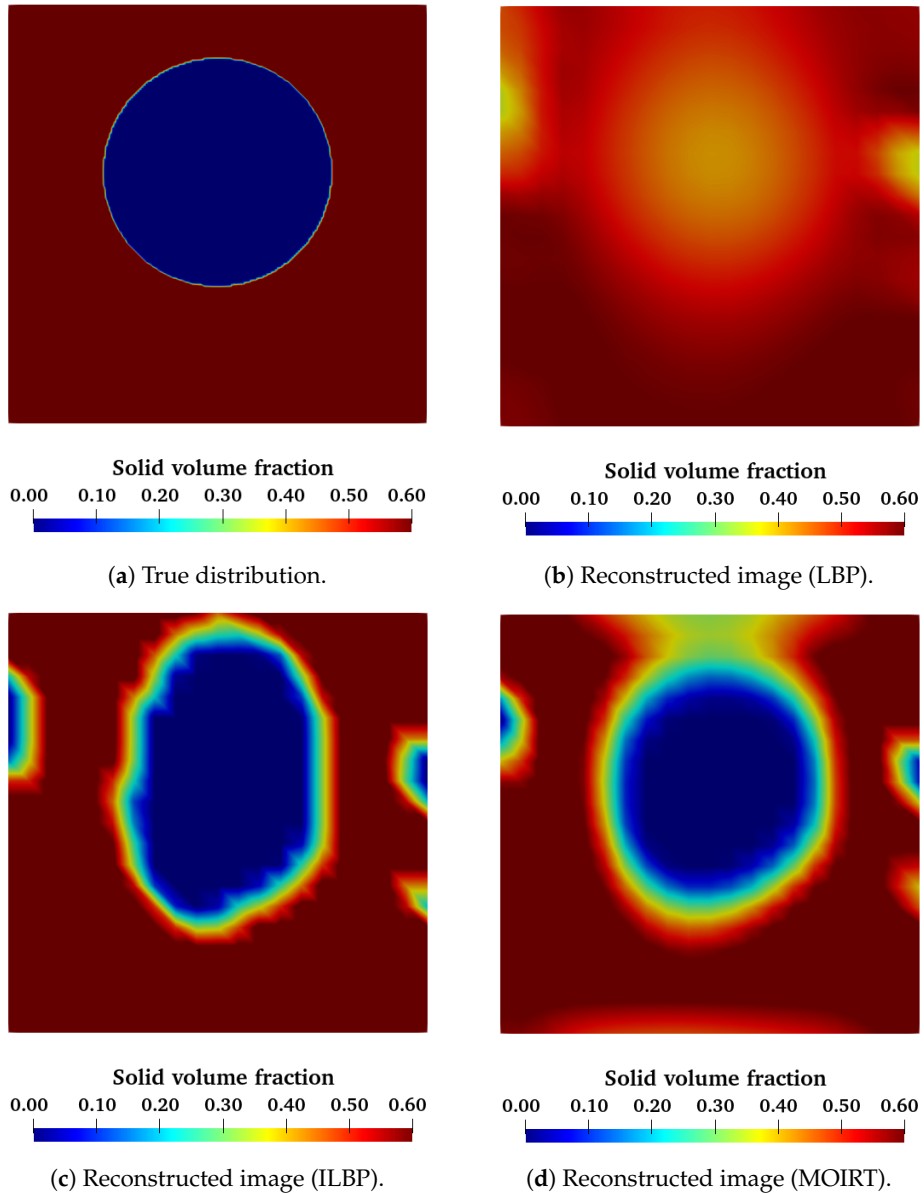

(**a**) True distribution.

(**b**) Reconstructed image (LBP).

(**c**) Reconstructed image (ILBP).

(**d**) Reconstructed image (MOIRT).

**Figure 2.** Comparison between the expected solid volume fraction distribution and the reconstructed solid volume fraction distribution for a void sphere of $d_{object} = 55$ mm using the classical reconstruction algorithms.

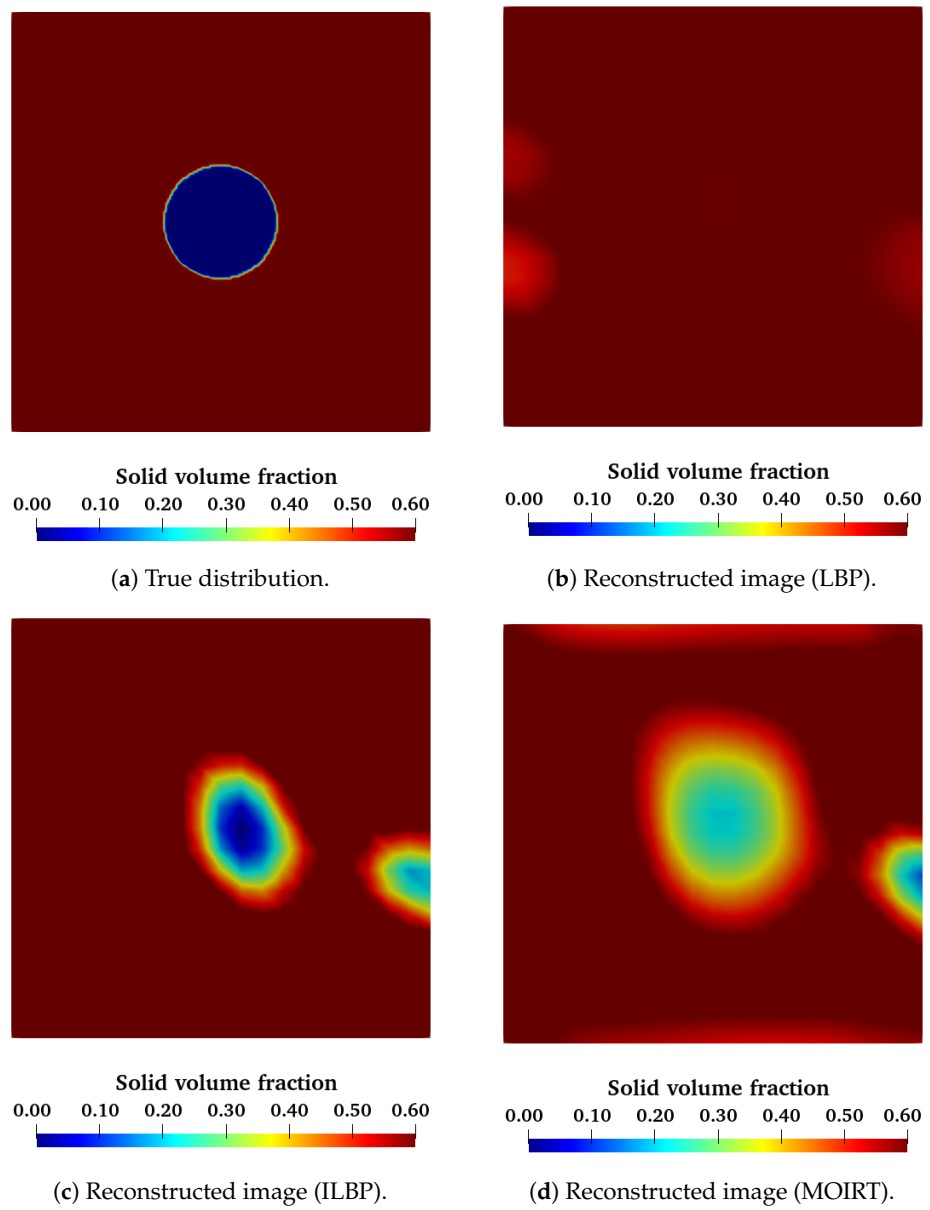

(**a**) True distribution.

(**b**) Reconstructed image (LBP).

(**c**) Reconstructed image (ILBP).

(**d**) Reconstructed image (MOIRT).

**Figure 3.** Comparison between the expected solid volume fraction distribution and the reconstructed solid volume fraction distribution for a void sphere of $d_{object} = 27$ mm using the classical reconstruction algorithms.

## 3. Artificial Neural Network

An ANN is a set of nodes, called neurons, organized in inter-connected layers (Figure 4). In a feed-forward ANN, a signal transits through the network from the input layer to the output layer. Each connection between two neurons, $i$ and $j$, has a weight, $w_{ij}$, associated with it. A single neuron receives the inputs from all the neurons in the previous layer and then computes the sum of all these values. After this, the final results are passed through a predefined activation function. The result from that activation function is the output of the neuron that is propagated into the next layer (Figure 5).

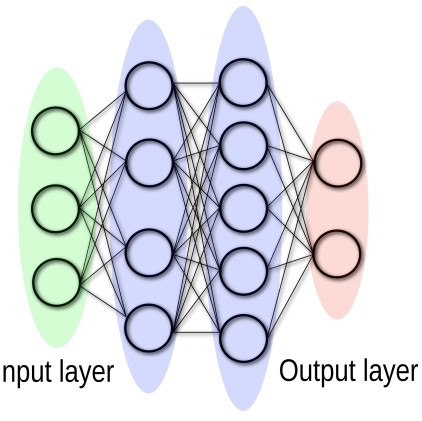

**Figure 4.** Illustration of a feed-forward artificial neural network with one input layer, two hidden layers, and one output layer.

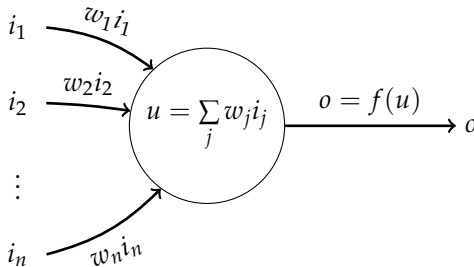

**Figure 5.** Illustration of how a single neuron works in an artificial neural network.

In supervised learning, the network must be trained so that the last layer outputs a desired signal for a given input signal in the first layer. To do this, a training database containing pairs of input and output vectors of values is used. An input vector is propagated through the network, and the error between the observed and expected output vector is measured via a loss function. The weights of the networks are subsequently modified by back-propagation so that the loss function is minimized.

Recently, ANN-based approach algorithms have been applied to the tomography image reconstruction problem. In the medical field, there have already been some attempts to obtain more accurate images for different imaging techniques. PET [12], X-ray CT [13,14], MRI [15], and SPECT [16] are some of the systems that have explored the pertinence of deep learning algorithms for the image reconstruction problem. The results show that a deep learning approach might be suitable for this type of problem and, in some cases, even outperforms the classical reconstruction algorithms.

In recent years, some efforts have been made toward the use of deep learning strategies for the image reconstruction problem in 2D ECT systems. Some approaches have used the raw capacitance data to try to predict the key hydrodynamics parameters: average volume fraction, flow patterns, and bubble diameter [17–19]. These studies, however, did not tackle the image reconstruction problem directly.

Recently, some works have been conducted to use artificial neural networks to address the image reconstruction problem of ECT devices. The ANN takes the capacitance measurements as input, and it aims to predict 2D volume fraction distribution. This technique has been very successful, with results that can be compared with the most performant algorithm already found in the literature. As we mentioned, machine learning algorithms require a training database from which to learn. In our case, this database should be composed of pairs of capacitance measurements and their corresponding solid volume fraction. Nevertheless, previous studies on the topic do not offer a satisfactory methodology to build this training database. Some works have proposed the use of images reconstructed

with classical algorithms [7]. However, using this approach, the ANN risks learning the shortcomings of the algorithm used. Another approach is to build a piece of software that creates random 2D volume fraction distributions resembling the patterns found in liquid-gas systems (stratified, annular, and core flows) [8]. Then, using electrodynamics simulation software, they calculated the capacitance values associated with the volume fraction distribution. The main drawback of this approach is that generating random volume fraction distribution for fluidized bed applications is much more difficult. Unlike the liquid-gas patterns, the 3D solid volume fraction distribution inside a fluidized bed is much more complex and unpredictable. Moreover, we would need an additional piece of software as well as knowledge to perform the electrodynamics simulations.

In the following sections, we will propose two different strategies to generate a training database to train an artificial neural network that can be easily implemented in any ECVT device if the sensitivity matrix is known: supervised learning and reinforcement learning. The choice of keeping the sensitivity matrix approach makes our strategy much simpler, and it is also based on the fact that classical reconstruction algorithms can achieve very good results even with the simplification of the sensitivity matrix.

### 3.1. CFD-Generated Training Database

The first strategy we used to generate the training data required us to train our ANN based on computer simulations; this is supervised learning. Previous works [8] have already built and trained ANNs using computer-generated data. However, their strategy consisted of generating artificial data that looked like the patterns found inside a two-phase flow (annular flow, stratified flow, single bar, and two bars). This allows us to generate a large number of different images to train our neural network. Nevertheless, we might not be able to accurately generate images corresponding to a 3D fluidized bed because the flow patterns are much more complex.

Nowadays, many CFD packages are very capable of accurately reproducing the general behavior of many different flow configurations (combustion, gas-solid flow, porous systems, etc.). These simulations give us information about the instantaneous solid volume fraction as a function of time. This information, combined with the sensitivity matrix, could allow us to simulate the capacitance measures associated with such distribution (Equation (1)). With the simulated solid volume fraction distribution and the simulated capacitance measurements, we can build an input/output database that can be used in the training process of an artificial neural network (Figures 6 and 7). An advantage of this method is that it can be easily applied to other fields that also have accurate simulation tools.

In this work, we will be focused on fluidized bed reactors. These systems can present different regimes depending on the types of particles and the fluidization velocity [20]. Difference regimes present different hydrodynamic and thermal properties, and identifying them correctly is very important for many industrial applications. Especially the transition between the bubbling regime and the turbulent regime [21,22]. However, the turbulent regime is characterized by very chaotic and complex behavior, particularly for solid volume fraction distribution. It is for these reasons that a training database consisting of only simple and structured patterns might not be suitable for an ECVT system used in a fluidized bed. However, the current CFD software and mathematical models allow for the accurate prediction of these complex regimes. Therefore, we could use this information to build our training database for an artificial neural network.

In order to simulate our fluidized bed reactor, we used the CFD software neptune_cfd. This is a multiphase Euler fluid code developed in the framework of the neptune project, financially supported by CEA, EDF, IRSN, and Framatome. It is capable of solving particle-laden flow problems in complex geometries using structured and non-structured meshes. This code has been extensively validated for a fluidized bed configuration using very accurate experimental techniques, such as positron emission particle tracking (PEPT) and radioactive particle tracking (RPT) [23–26]. neptune_cfd is a massively parallel code [27]; this allows for the obtainment of a large training database very quickly. The simulated

geometry is a column 10 cm in internal diameter and 1 m in height in the Z direction. We used an O-grid mesh with 400,000 cells, approximately 3 mm in length. For the solid phase, we used glass beads 250 μm in diameter with a density equal to 2700 kg/m$^3$. The gas phase is air at 20 °C and atmospheric pressure. The initial height of the fluidized bed was set to 17 cm. The interested reader can consult [24] for more details on the mathematical models used by neptune_cfd.

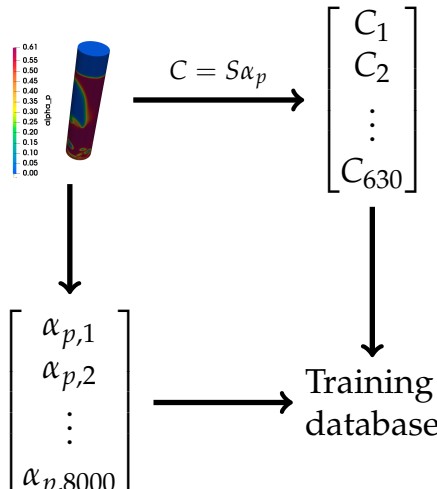

**Figure 6.** Strategy used to generate the training database for the artificial neural network.

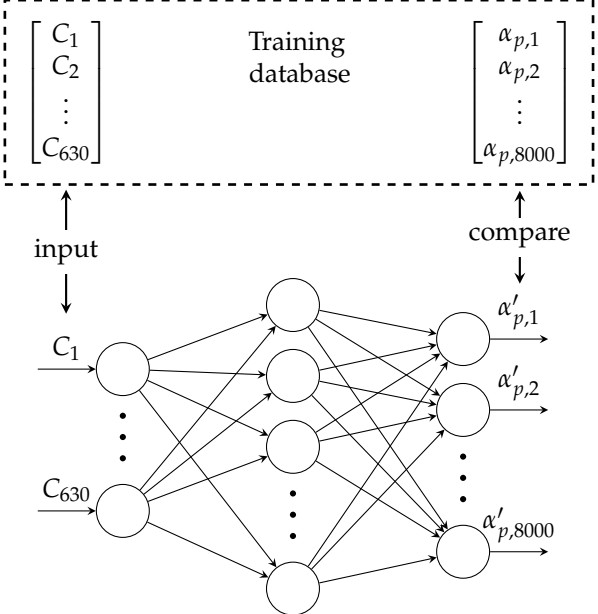

**Figure 7.** Scheme of the training strategy using CFD-generated data (the ANN structure is not representative of the ANN used).

As we wanted to obtain the most diverse database possible, we used different gas velocities at the inlet. We performed simulations for four different inlet velocities ($3U_{mf}$, $4U_{mf}$, $6U_{mf}$, and $7U_{mf}$), where $U_{mf}$ = 6.3 cm/s is the minimum fluidization velocity obtained using Ergun's correlation [28]. For each simulation, we recorded the solid volume fraction distribution in the 8000 voxels corresponding to the exact location of the voxels reconstructed by the ECVT system. For each case, we simulated 30 s of physical time. We extracted the solid volume fraction inside a 10 cm height region between $h_1$ = 6 cm and $h_2$ = 16 cm, where $h$ = 0 cm represents the column inlet. The acquisition frequency was set

to 100 images per second. Given that the expected bubble frequency is about 1.5–2.5 Hz, this acquisition frequency is high enough to capture the bubbles passing through the sensing region. The final database is, therefore, composed of 12,000 pairs of capacitance/solid volume fraction vectors. The size of this database was chosen after a sensitivity analysis of the results by varying the size of the training database.

In order to build our ANN, we used Keras. This is a open source Python library designed to build artificial neural networks quickly [29]. In the back end, Keras uses the symbolic mathematical library TensorFlow [30].

The artificial neural network used is composed of an input layer of 630 neurons (corresponding to the 630 capacitance measurements) followed by three hidden layers of 1024, 2048, and 4096 neurons, respectively, and a final output layer of 8000 neurons corresponding to the 8000 values of solid volume fraction. The choice of these parameters was made after analyzing different configurations with different numbers of hidden layers and numbers of neurons per hidden layer. This final configuration was chosen as it was the most performant ANN for the supervised learning approach and the reinforcement learning approach presented later. In order to ensure that the solid volume fraction predicted by the neural network is bounded between 0 and 0.64 (the solid volume fraction at maximum packing), we enforced a scaled sigmoid activation function for the output layer of the ANN:

$$\alpha_{p,i} = 0.64 \frac{1}{1 + e^{-u_i}} \tag{3}$$

where $u_i$ is the sum of all inputs of the $i$-th neuron in the output layer.

Before the training phase, we shuffled the training database so two consecutive entries were not correlated. We also randomly removed 25% of the entries of the training database for validation purposes. In this way, our ANN was trained with 75% of the available data, and the remaining 25% was used to quantify the precision of the algorithm during the training. For the training phase, there are two key parameters to specify: the batch size and the number of epochs. The first one refers to the number of samples of the training database that will be used to compute the loss function before updating the ANN weights. For example, if our training database consists of 9000 samples, and if we choose a batch size equal to 20, first, the entries 1–20 in the training database will be fed to our neural network, and the loss function of these 20 entries will be used to change the weights, $w_{ij}$. Then, we will feed the entries 21–40 to the updated ANN, and the new loss function value is used to change the weight values. This process is repeated until all 9000 entries are used. A small batch size could allow us to train the database faster because the weights are updated more frequently. However, choosing a very small batch size could generate important fluctuations that could harm the convergence rate. The second parameter that has to be specified is the number of epochs. The epochs are the number of times the whole database is used during the training phase. If the number of epochs is set to 10, this means that the learning algorithm will go over the 9000 samples 10 times. A high number of epochs can improve the quality of the ANN, but this could also be very time-consuming. In our example, we chose the batch size to be equal to 20, and we trained over 600 epochs.

In order to monitor the convergence of the ANN, we can calculate the root mean squared error (RMSE) between the predicted ($\alpha'_p$) and the expected ($\alpha_p$) solid volume fraction values (Equation (4)).

$$\text{RMSE}_{\alpha_p} = \sqrt{\frac{1}{n} \sum_{i=1}^{n} \left( \alpha_{p,i} - \alpha'_{p,i} \right)^2} \tag{4}$$

where $n$ is the total number of voxels in the sensing region.

In Figure 8, we can see the RMSE during the training phase when using the training and validation databases. As we can observe, the RMSE decreases as the number of epochs increases for the training database. This means that for each epoch, the output predicted by the neural network is closer to the expected output. We observe the same trend when we evaluate the accuracy of the neural network using the validation database (which is

not used to update the internal weights of the ANN). This highlights that the ANN is also capable of predicting good values for inputs that are not present in the training database. This is a good sign because it shows that the neural network can also provide accurate results for data outside the training database (unseen data). After 600 epochs, we see that RMSE is decreasing slowly, which means that the training algorithm has converged, and the reconstructed images do not change if we keep training the ANN. Now, we can take the neural network obtained in the last epoch and use it to reconstruct images using new data.

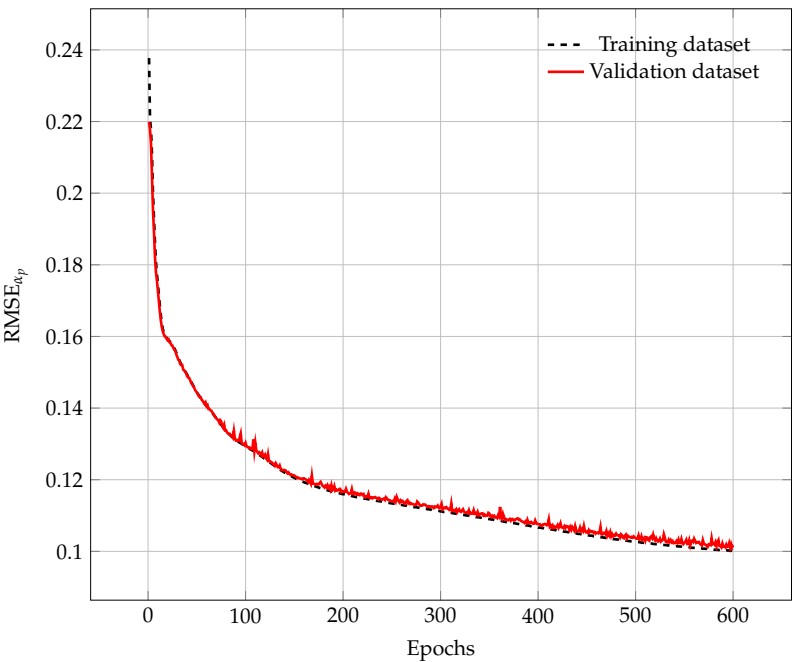

**Figure 8.** Root mean square error as a function of the training epochs.

In order to evaluate the quality of our trained ANN, we can feed it with new simulated data generated using a different inlet velocity than the ones used during the training phase ($5U_{mf}$). For this simulation, we can extract the solid volume fraction and the capacitance values, as we did to construct the training database. Then, we pass the capacitance values as input to our ANN, and we compare the output solid volume fraction distribution with the simulation results. In Figure 9, we compare the simulation output with the reconstructed image from the ANN for a 2D slice XZ plane in the middle of the column at three different moments in time. The first image corresponds to the first moments after fluidization started, where we have two symmetrical bubbles rising. The second image is the moment where a big air bubble rises, and the third image corresponds to a more complex structure appearing in the reactor. We remark that for all three cases, the reconstructed images are very close to the output of the simulation. These images show that the ANN is capable of reconstructing the 3D representation of a fluidized bed using simulated capacitance data as well as the global topology of the flow.

We can also compare the RMSE between the reconstructed images and the simulation for each time step (Figure 10). For ANN, the black line represents the RMSE as a function of time, while the dashed blue line is its mean value over time. As a comparison, we also drew (in dashed lines) the mean RMSE of the images obtained with the classical reconstruction algorithms. This shows that the ANN is very close to the most performant algorithms. If we compute the mean absolute error between the simulated values and the predicted values, we get that the ANN predictions have an error in $\alpha_p$ of 0.06. Given that the average $\alpha_p$ value in a fluidized bed is of the order of 0.40, our ANN model has a prediction error of 15%. This error is low enough to allow us to capture the large structures developing inside the reactor accurately.

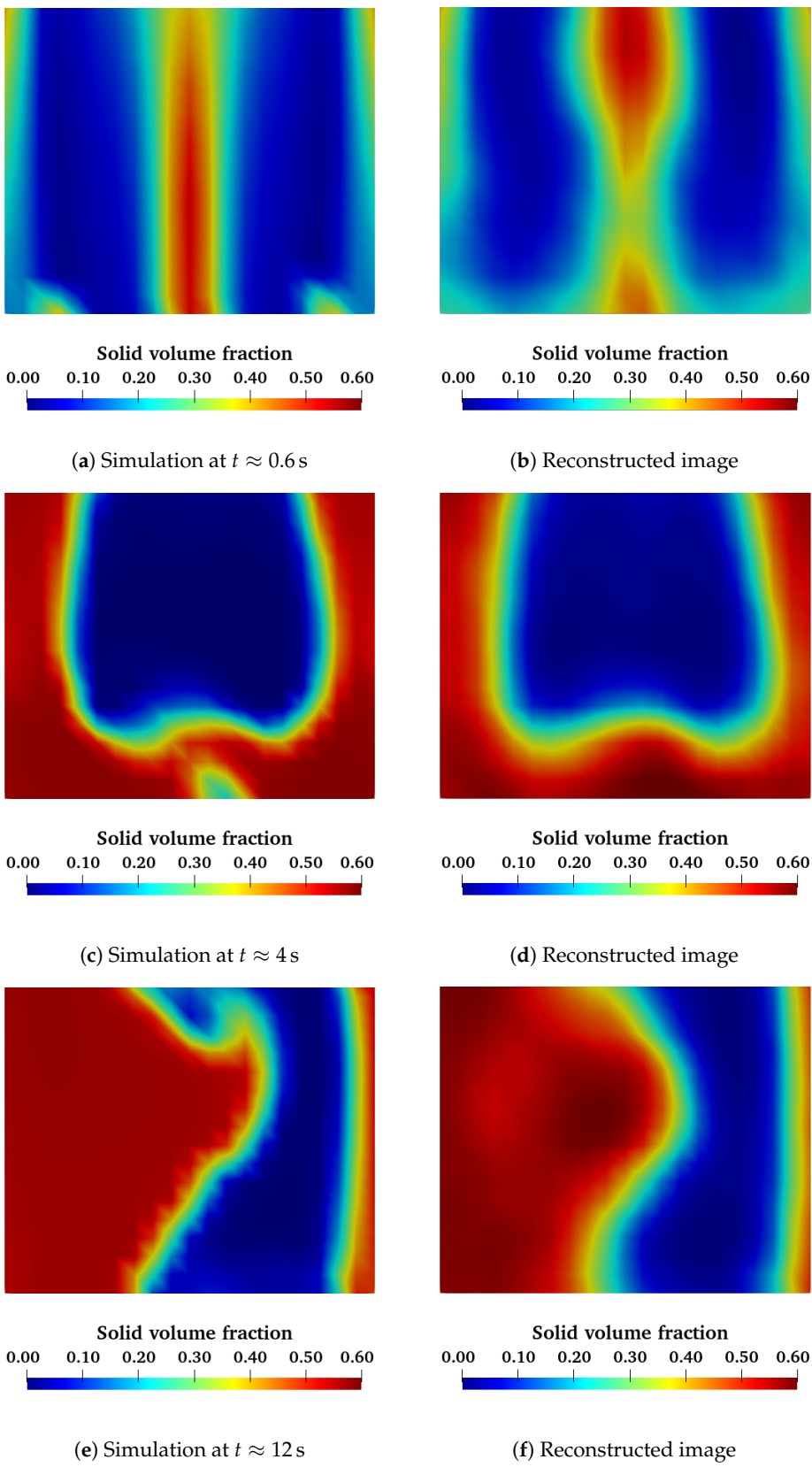

(**a**) Simulation at $t \approx 0.6\,\mathrm{s}$

(**b**) Reconstructed image

(**c**) Simulation at $t \approx 4\,\mathrm{s}$

(**d**) Reconstructed image

(**e**) Simulation at $t \approx 12\,\mathrm{s}$

(**f**) Reconstructed image

**Figure 9.** Comparison of some instantaneous solid volume fraction distributions between the numerical simulations (**left**) and the reconstructed image using an ANN trained with CFD data (**right**) (slice in the XZ plane).

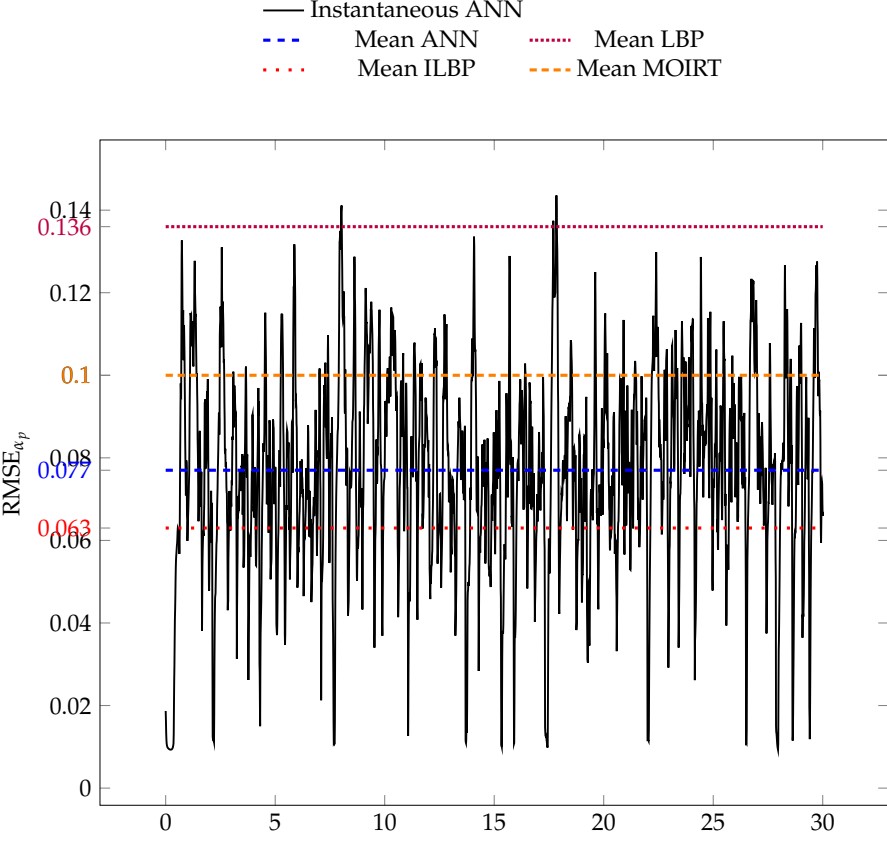

**Figure 10.** Root mean square error as a function of time between the numerical simulation and the image reconstructed by the ANN. The blue dashed line represents the time average, and the rest of the colored dashed lines represent the time average of the RMSE using the classical algorithms.

We can also test this neural network using the experimental data obtained with the void spheres in a fixed bed of glass beads. Figure 11 shows a slice of the reconstructed volume for both the small and the big spheres. These images show that the ANN was able to detect a spherical object inside the volume. As with the classical algorithms, the big sphere is much easier to recognize than the small sphere. We can characterize the diameter of all test spheres using the same criteria as before (see Table 2). As we can see, we have an overestimation for every object. However, these values are still close to the real expected value.

**Table 2.** Equivalent diameter of the reconstructed objects using the classical algorithms and an artificial neural network trained using a supervised learning technique.

| $d_{obj}$ | 55 mm | 50 mm | 44 mm | 40 mm | 27 mm |
|-----------|-------|-------|-------|-------|-------|
| LBP | × | × | × | × | × |
| ILBP | 57 mm | 53 mm | 49 mm | 42 mm | 18 mm |
| MOIRT | 56 mm | 54 mm | 49 mm | 43 mm | 23 mm |
| ANN | 60 mm | 59 mm | 55 mm | 50 mm | 31 mm |

Once the neural network has been trained, the reconstruction process is very fast because it only requires propagating the capacitance signal through the network. This makes this strategy much more efficient than the classical iterative algorithms. However, the training phase can be computationally expensive depending on the size and complexity of the network, the number of training epochs, and the size of the training database. Our neural network took around 10 h to perform the entire training phase for the same

computing power described above. Nevertheless, this neural network can reconstruct one image in around 45 ms. This makes this strategy as fast as the fastest algorithm, LBP, and as accurate as the most complex algorithm (MOIRT) but much faster.

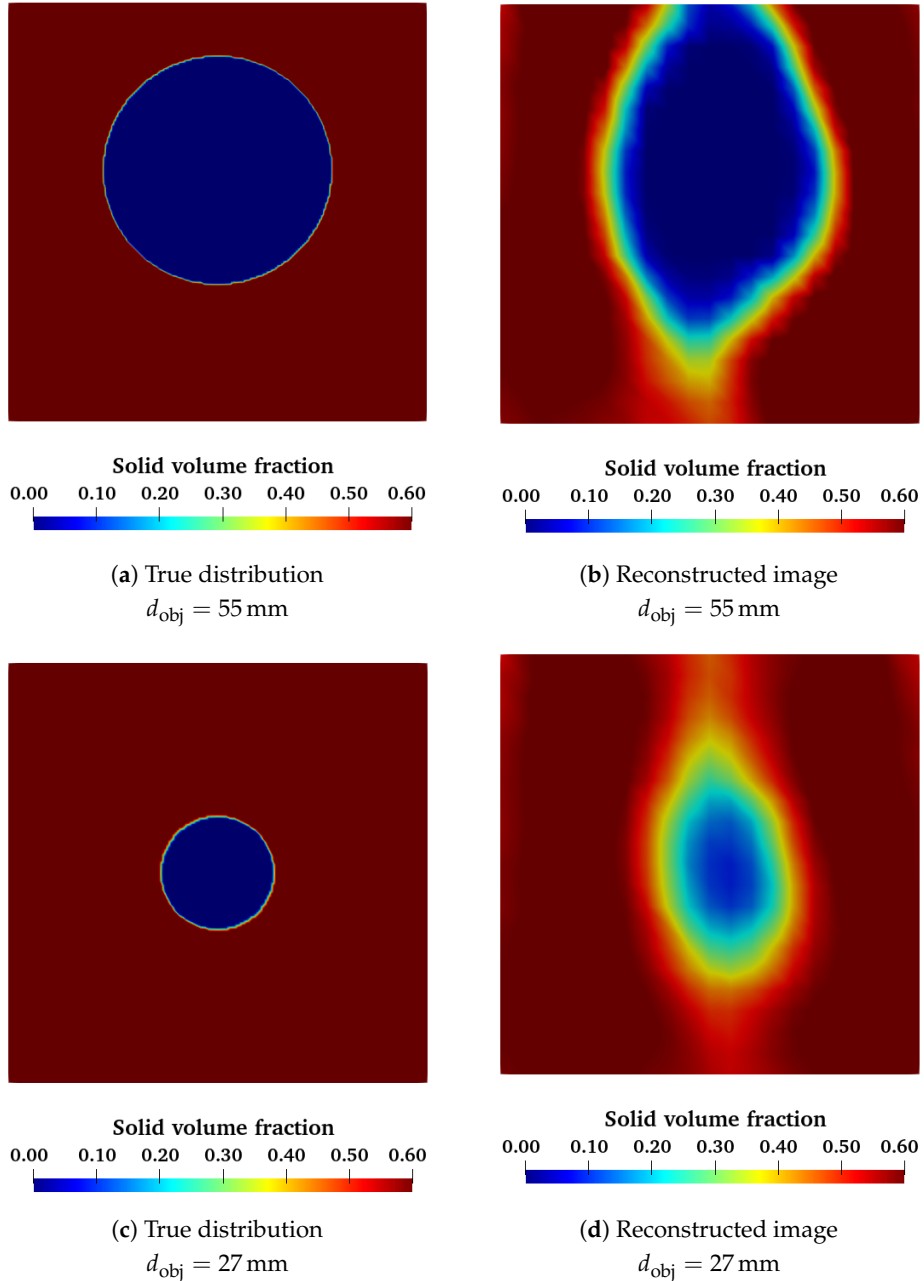

**Figure 11.** Comparison between the expected solid volume fraction distribution and the reconstructed solid volume fraction distribution using an ANN trained with CFD data for a void sphere of $d_{\text{obj}} = 55$ mm and $d_{\text{obj}} = 27$ mm (2D slice in the XZ plane).

This makes this approach suitable for a system where accurate and instantaneous post-processing is needed, or a large amount of data needs to be processed. However, an important drawback is the necessity of a way to generate simulated data. This means that this approach depends on other simulation tools. In addition to this, we need to make sure that the simulated data represents the physical phenomenon accurately; any bias or error present in the simulated data could also be reproduced by the neural network. In the next section, we present a different approach to making a standalone artificial neural network.

### 3.2. Experimental Generated Training Database

We propose a second strategy to build the training database needed for the training phase of the ANN: reinforcement learning. In this approach, we no longer need a database composed of input/outputs ($\mathbf{C}/\alpha_p$). Instead, we can directly use experimental data even without any previous knowledge of the solid volume distribution.

The key aspect of this approach is that we know how to estimate the capacitance measurements given a solid volume fraction distribution (Equation (1)). During the training phase, we can feed experimental data, $C$, directly into the ANN. Instead of comparing the AAN's output $\alpha'_p$ to some true $\alpha_p$ distribution, we are going to use Equation (1) to transform our predicted $\alpha'_p$ into predicted capacitance values $C'$. If the neural network is well-trained, the values $C$ and $C'$ must be similar (Figure 12). This will mean that the generated 3D image corresponds to the original input capacitance data. If this is not the case, then the internal weights have to be adjusted. Hence, our neural network will be trained so it minimizes the RMSE between $C$ and $C'$ (Equation (5)).

$$\text{RMSE}_C = \sqrt{\frac{1}{m}\sum_{i=1}^{m}\left(C_i - C'_i\right)^2} \tag{5}$$

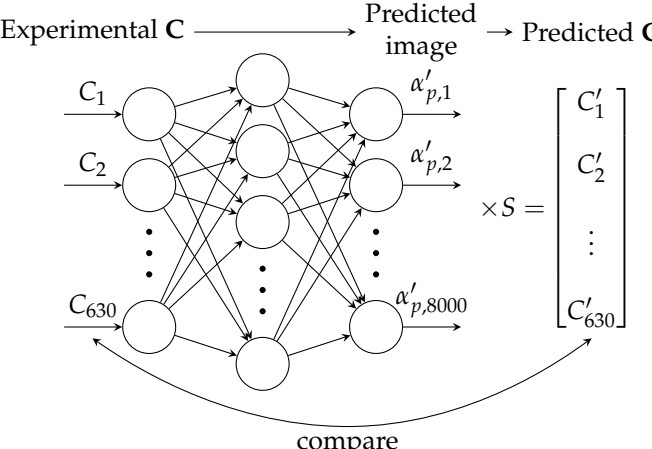

**Figure 12.** Training strategy using a training database with only experimental data.

This approach has the advantage of not needing computer-simulated data to train itself. We can now directly use experimental data, even without having previous knowledge of the true solid volume fraction distribution. This makes this technique completely independent of any external tool. Another advantage is that we can use any new experimental data to simultaneously reconstruct the solid volume fraction and to train the neural network even further, which is in contrast to the previous approach, where the experimental data cannot be used to improve the neural network. Therefore, this technique allows us to have a self-sufficient ANN that can be in a constant learning process.

In order to test this approach, we built the same neural network used in the previous section—an input layer of 630 neurons, three hidden layers of 1024, 2048, and 4096 neurons, and one output layer with 8000 neurons—with the same sigmoid activation function used before (Equation (3)). In order to obtain the experimental data, we placed our ECVT electrodes around a plexiglas column 10 cm in internal diameter and 1 m in height (see Figure 13). These are the exact same dimensions of the geometry used in the numerical simulations. The ECVT device is placed at the same location as in the simulation, between $z = 6$ cm and $z = 16$ cm. The solid phase is composed of glass beads 250 μm in mean diameter. The gas phase is air at ambient pressure and ambient temperature, with 50% relative humidity, to ensure that the electrostatic effects do not appear in the solid phase. Similar to the previous approach, we used different inlet velocities between $3U_{mf}$ and $7U_{mf}$.

The bottom of the ECVT system was placed 6 cm above the inlet to match the settings used in our simulations exactly. The acquisition frequency was set to 50 frames per second, and we took 3000 frames for each inlet velocity. We also used the exact training parameters for this ANN, with a batch size equal to 20, and we trained the neural network over 600 epochs. Figure 14 shows the evolution of the RMSE as a function of the training epochs. We remark that the RMSE decreases when the number of training epochs increases. This means that the neural network is converging to a better solution. This behavior is true for both the training and the validation databases. This shows that this ANN can also be used for data that were not present in the training database.

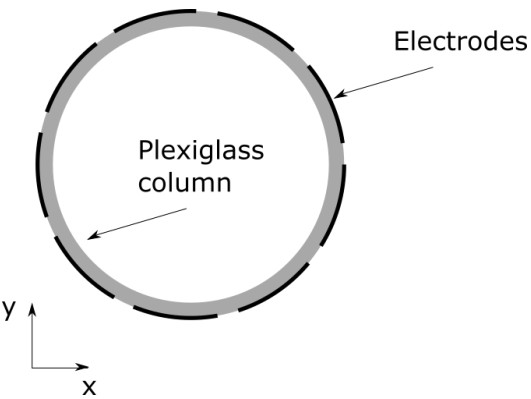

**Figure 13.** Position of the ECVT electrodes with respect to the experimental fluidized bed.

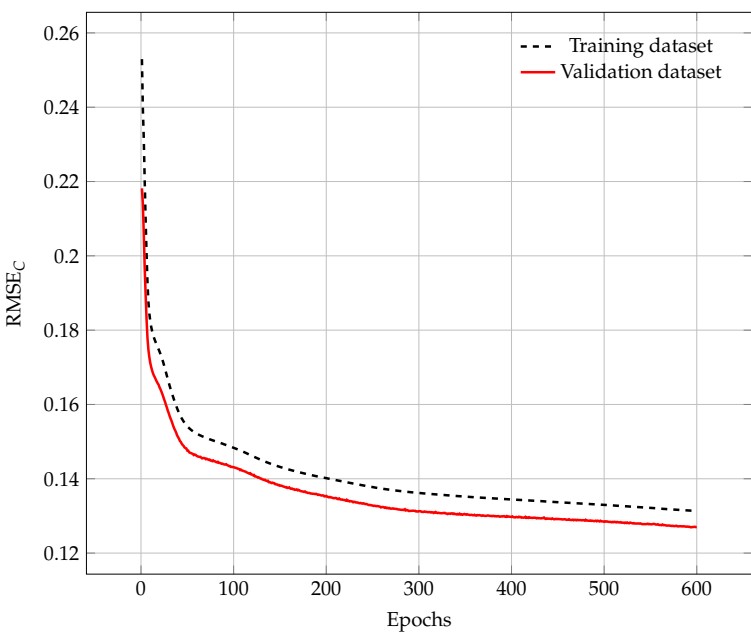

**Figure 14.** Root mean square error as a function of the training epochs.

With this trained artificial neural network, we can perform the same analysis as we did for the previous one. First, we can feed our ANN with simulated data and compare the reconstructed images with the numerical simulation. In Figure 15, we can see that, qualitatively, the results produced by this approach are in good agreement with the expected results. However, they are not as good as the results produced by the CFD-trained ANN. The two symmetrical bubbles rising at the start of the simulation are not well captured (Figure 15a,b). For the big bubble (Figure 15c,d) and the complex structure near the wall (Figure 15e,f), we obtained a more accurate reconstruction. Nevertheless, the results are worse than the previous ANN strategy.

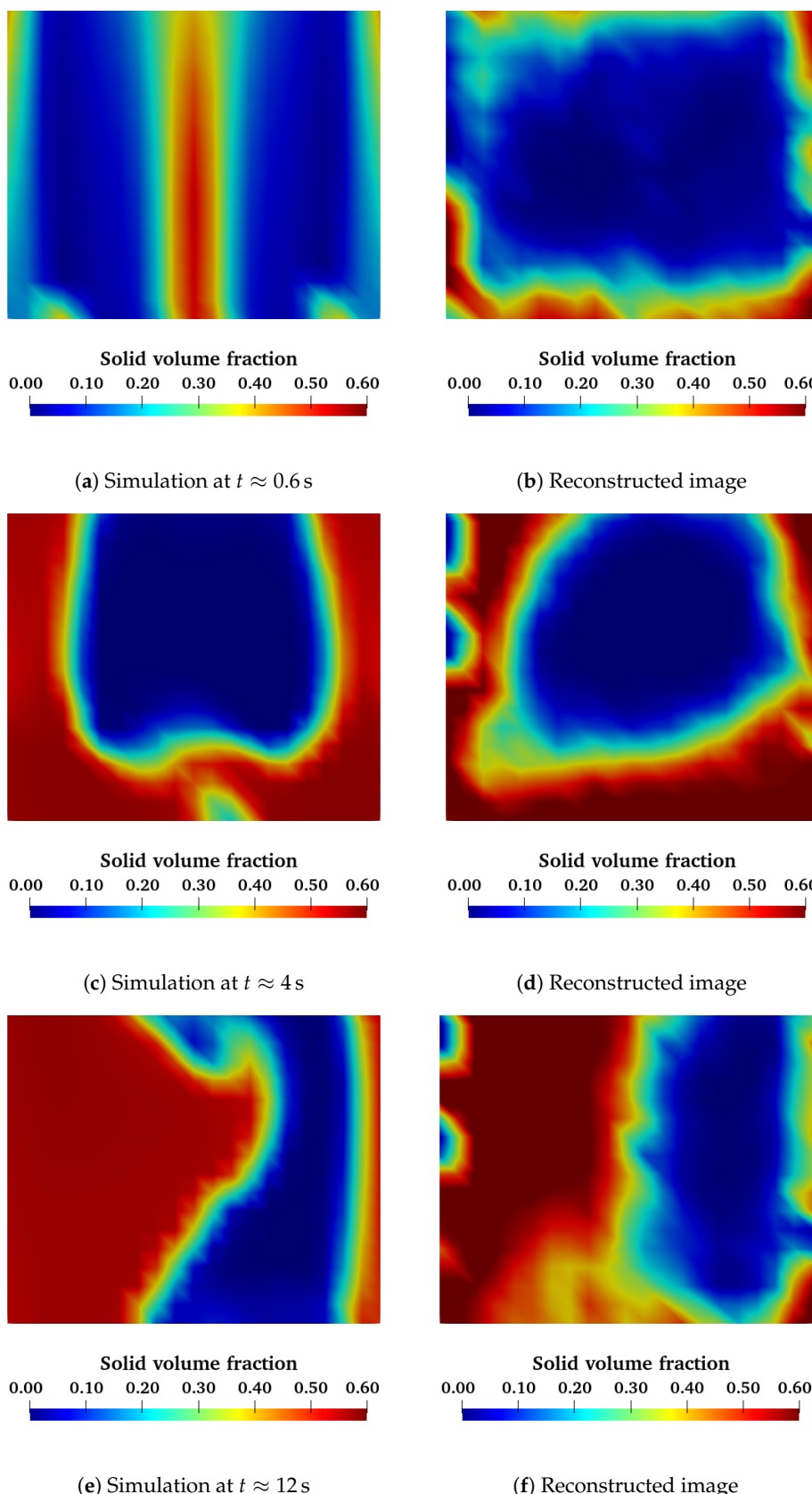

**Figure 15.** Comparison of some instantaneous solid volume fraction distributions between the numerical simulations (**left**) and the reconstructed image using an ANN trained with experimental data (**right**) (slice in the XZ plane).

Figure 16 represents the root mean squared error between the numerical simulation and the reconstructed image as a function of the simulation time. We can see that the mean RMSE value is around 0.13, which is worse than the previous ANN. This curve also shows that this ANN is also not as good as the ILBP and MOIRT algorithms. However, it does perform better than the LBP scheme. For this approach, the mean absolute error between the predicted and the expected solid volume fraction values is equal to 0.08. This means that this approach has an error of 22% compared to the mean solid volume fraction found in a fluidized bed.

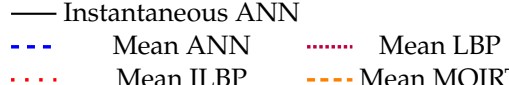

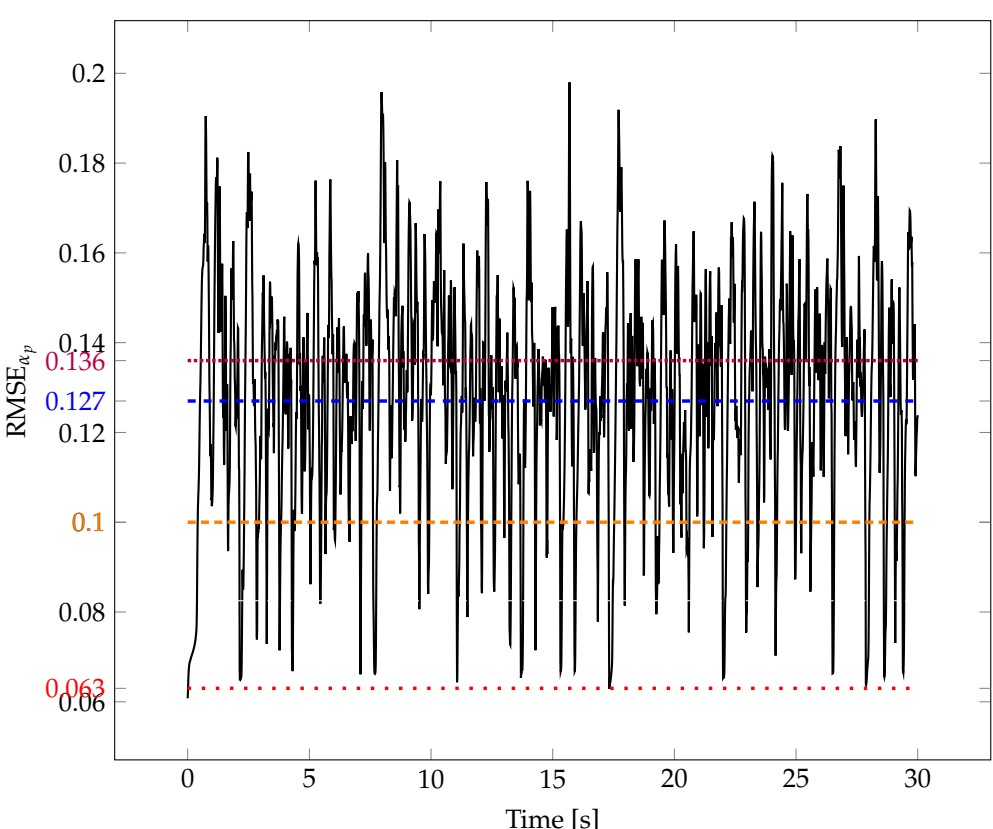

**Figure 16.** Root mean square error as a function of time between the numerical simulation and the image reconstructed by the ANN. The blue dashed line represents the time average, and the rest of the colored dashed lines represent the time average of the RMSE using the classical algorithms.

This ANN can also be used to compare against the experimental data obtained with the void spheres in the fixed bed. A visual comparison is made in Figures 17 and 18. These images reveal that our ANN has no problem detecting big void objects inside a fixed bed of glass particles. It does, however, fail to reconstruct the smallest of our test objects. The equivalent diameters of the reconstructed objects are shown in Table 3. We can see that the equivalent diameters are very close to the real diameters for large objects, but smaller objects are not detected at all. This shows that for a similar training database and network architecture, the reinforcement approach does not perform as well as the supervised approach.

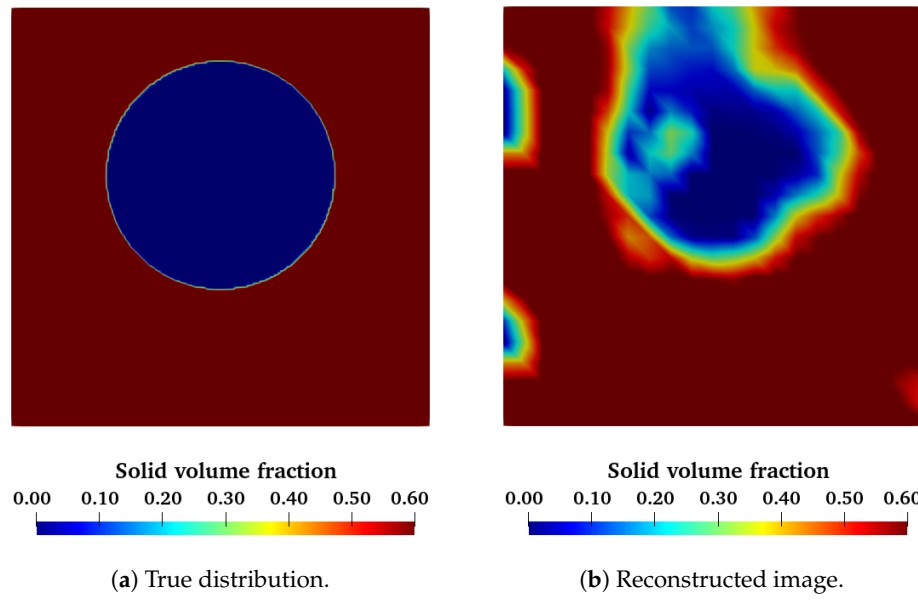

(**a**) True distribution.　　　　　(**b**) Reconstructed image.

**Figure 17.** Comparison between the expected solid volume fraction distribution and the reconstructed solid volume fraction distribution using an ANN trained with experimental data for a void sphere of $d_{\text{obj}} = 55\,\text{mm}$ (2D slice in the XZ plane).

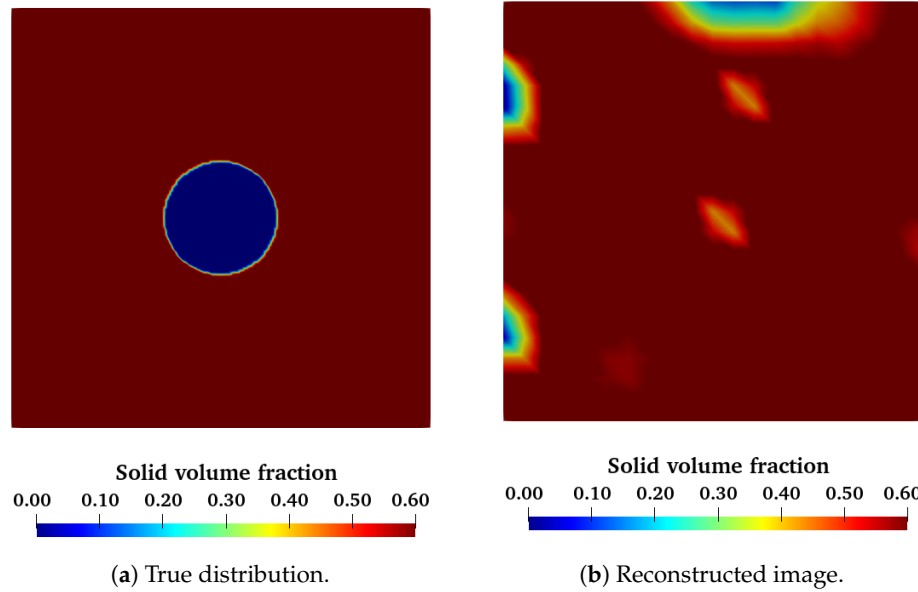

(**a**) True distribution.　　　　　(**b**) Reconstructed image.

**Figure 18.** Comparison between the expected solid volume fraction distribution and the reconstructed solid volume fraction distribution using an ANN trained with experimental data for a void sphere of $d_{\text{obj}} = 27\,\text{mm}$ (2D slice in the XZ plane).

**Table 3.** Equivalent diameter of the reconstructed objects using the classical algorithms and an artificial neural network trained using a supervised learning technique (SL) and the reinforcement learning technique (RL).

| $d_{\text{obj}}$ | 55 mm | 50 mm | 44 mm | 40 mm | 27 mm |
|---|---|---|---|---|---|
| LBP | × | × | × | × | × |
| ILBP | 57 mm | 53 mm | 49 mm | 42 mm | 18 mm |
| MOIRT | 56 mm | 54 mm | 49 mm | 43 mm | 23 mm |
| ANN-SL | 60 mm | 59 mm | 55 mm | 50 mm | 31 mm |
| ANN-RL | 56 mm | 53 mm | 43 mm | × | × |

The computational cost of using this neural network is almost the same as the previous one. The training phase took around 10 h, and the reconstruction of one image takes about 50 ms. Despite the fact that we obtained a less performant ANN, this strategy showed some promising results. We are able to reconstruct simulated data with acceptable spatial accuracy. Moreover, the algorithm is capable of detecting big objects inside the sensing regions. However, further research is needed to improve the network for capturing small objects.

These results prove that we can train an artificial neural network using solely experimental data in order to address the image reconstruction problem for 3D ECVT systems. Our ANN was tested against both simulated and real data. The results highlight that this approach is also suitable as an image reconstruction algorithm. Although the results were slightly worse compared to the first strategy, this approach is completely self-sufficient and does not need any external tool. This strategy will also benefit from any new experimental data to learn and increase its quality.

Despite the fact that this training technique did not perform as well as the previous one, it is worth noting that there are some fundamental differences between these two approaches. The CFD data used to train our first artificial neural network was totally free from noise, while the experimental data are always convoluted with the noise generated by the acquisition configuration. At this stage, it is unknown if this noise has an effect on the quality of the training process. This is something that should be taken into account when working with this approach; however, it is unclear how this effect could be modeled. Despite this, reinforcement learning has shown promising results and should be further investigated.

It is also worth noting that both approaches gave results that are of the same order of magnitude as the error obtained using the classical reconstruction algorithms. This might be due to the use of the sensitivity matrix approximation to solve the forward problem (Equation (1)), which is the core of classical reconstruction algorithms and also our ANN-based approach. This approximation assumes that the sensitivity distribution of the ECVT is independent of solid volume fraction distribution. This hypothesis is not true in electric tomography. However, the sensitivity matrix approach has proven to be a valid strategy to tackle the reconstruction problem, both for the classical reconstruction algorithms and, now, for an ANN-based approach. Nevertheless, more accurate results are to be expected if the exact formulation of a forward problem is incorporated into the methodology. This would, however, add an extra layer of complexity.

## 4. Conclusions

In this work, we present two different strategies to build a training database for a machine learning-based algorithm for ECVT systems applied to fluidized beds. The first proposition is based on accurate 3D numerical simulations. From these simulations, we can extract the solid volume fraction distribution in different regimes and conditions, and we can deduce the capacitance measurements using the sensitivity matrix model. These data allowed us to train an artificial neural network that tackles the image reconstruction problem in ECVT devices. First, we visually compared the reconstructed images generated by the ANN to the CFD simulations. The results obtained showed excellent agreement. The ANN was even able to reconstruct the most complex patterns present in a fluidized bed. We also performed a quantitative analysis comparing the error of the ANN approach to the different classical algorithms found in the literature. Our study showed that the artificial neural network was as accurate as the best algorithms found in the literature; however, the ANN can process data much faster.

The second approach is based on experimental data without needing previous knowledge of the solid volume fraction distribution. In this case, we aim to obtain a reconstructed image that corresponds as closely as possible to the input capacitance values obtained by using the sensitivity matrix approximation. In this way, we get a self-sufficient technique that does not depend on any other tool. This approach was trained using an equivalent

experimental database to the one generated for the first approach. Our results show that this methodology performs worse in the simulated data to be restored.

It is worth noting that both approaches rely heavily on the sensitivity matrix approximation to solve the forward problem. This is an important assumption that relies on the hypothesis that the sensitivity of the device is independent of the solid distribution inside the sensing region. A more accurate approach would be to use electrodynamics to solve the forward problem accurately. However, this would add an extra layer of complexity to the approach. Nevertheless, we have shown that the sensitivity matrix approach can still be coupled with an ANN approach to create a reconstruction algorithm.

For the sake of simplicity, this work only studied the most simple neural network configuration: a feedforward neural network. However, a more complex solution could be studied. Recurrent neural network models, such as LSTM [31], or attention models inspired by the recent transformer architecture [32] would allow for the exploitation of the sequential nature of the inputs, which is ignored with the feed-forward model. Another perspective of this work would be to use recurrent neural networks to predict the capacitance values at time $t$, knowing the values at times $[t_0..t_{-1}]$. These predictions could be used for capacitance captor failure detection in ECVT systems. Finally, the CFD-generated data were noiseless, while the experimental data were always convoluted with noise; this could explain some of the differences between the two approaches presented in this study. The impact of this factor on the training database and the predicted image quality also needs to be further investigated.

**Author Contributions:** Experimental set up and ECVT measures, C.M., E.C. and R.A.; CFD Modeling and data generation, C.M. and R.A.; ANN modeling and training strategy, A.M., R.N., D.S. and S.N.; ANN training, tests and validation, C.M., R.A., R.N., A.M., D.S. and S.N.; Results Analysis, C.M., E.C., R.A., D.S. and S.N.; Writing the original draft, C.M., R.A. and D.S.; Article review and editing, C.M., R.A., E.C. and S.N.; Project Administration, R.A. and S.N.; Funding Acquisition, R.A. All authors have read and agreed to the published version of the manuscript.

**Funding:** This work was supported by the ANR-IPAF project, grant ANR-16-CE06-0008 of the French National Agency of Research (ANR).

**Data Availability Statement:** Data are contained within the article.

**Conflicts of Interest:** The authors declare no conflict of interest. The funders had no role in the design of the study; in the collection, analyses, or interpretation of data; in the writing of the manuscript; or in the decision to publish the results.

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
