# Peer review of "On Using CFD and Experimental Data to Train an Artificial Neural Network to Reconstruct ECVT Images: Application for Fluidized Bed Reactors"

_processes, doi:10.3390/pr12020386_

Round 1
Reviewer 1 Report
Comments and Suggestions for Authors
This is a novel approach in generating the dataset for machine learning for ECVT image reconstruction. Fluidized bed are fairly important in chemical engineering applications. The work is significant, however, a few clarification are requested.
1. The title could be more descriptive in regard to the methodology of the manuscript.
2. A fundamental limitation of this work is the use of equation (1) for calculation of capacitance data as indicated in Fig 3. This should have been done with an electrodynamics code/software. Equation (1) is only an approximation. Because of this, the performance with experimental data is likely to be affected. Therefore, the authors are requested to provide a few reconstruction examples with experimental data, preferably with the supervised model.
3. The fact that the capacitance data is calculated using the sensitivity matrix only (equation 1, Fig 3) should be clarified either in the introduction, or in the conclusion.
4. Page 5, line 171: why this particular parameters are chosen for the ANN?
Comments on the Quality of English LanguagePlease correct a few typos found here and there.
Author Response
Please find in the file enclosed the authors reply to your benefit comments.
Thank you for helping us to improve our proposition.
Kind Regards

Reviewer 2 Report
Comments and Suggestions for Authors
I suggest rejecting the article with the option to resubmit. The methodology of scientific research is poor. Experiments require repetition. The article is missing a lot of information. False claims are being made.
The name "ECVT" is promoted by one research group (Warsito), although this name is incorrect and creates terminological confusion. If tomographic reconstruction uses 3D measurement data and generates 3D images, it is called 3D tomography. This nomenclature applies to every tomographic technique, from X-ray tomography to impedance tomography. Why should ECT be any different? An article using misleading nomenclature should not be published.
The authors claim that ECT is extensively used in monitoring processes in fluidized beds. Is this true? Some researchers claim that 3D ECT makes no sense due to the very small measurement values in the 3D configuration of the electrodes.
I think the title is misleading. We cannot talk about a database if we only have two training sets. The subject is the training database. Is this an open data database?
"Because they do not rely on an iterative algorithm to reconstruct the 3D image …". Do you really need 3D in this sentence? This is true also for 2D reconstruction.
"Previous works have used very limited or simple datasets to train their neural network …" There is no evidence that this is true. You provide no literature reference.
"This matrix is known because it only depends on the system geometry, the sensors characteristics and the desired resolution." That is not true in electric tomography! We do not know the sensitivity matrix from this linear approximation as it also depends on the unknown distribution in the reconstructed volume.
"For this reason, a new type of algorithm has been developed using machine learning techniques." There is no any citation. Are you repeating an approach already used in ECT?
A paragraph explaining what a neural network is is unnecessary in a scientific article. The same concerns Figures 1 and 2.
"The main drawback of this approach is that generating random volume fraction distribution for fluidized beds applications is much more difficult." Is this a problem? Do we really need CFD to simulate 3D distribution? If so, you do not show the evidence in your paper. Does your CFD software show individual solid particles?
"accurately reproducing the general behavior of many different flow configuration (combustion, gas-solid flow, porous systems, etc)" is an advantage, but this can lead to overtraining as you do not use any random distributions.
Figure 3. What is alpha-p? The 2D image is shown in the picture. How is it related to the 3D distribution?
"This frequency is high enough to capture the bubbles passing through the sensing region." You do not specify the size of the bubbles. Does your CFD software show individual solid particles? Are the solid particles in a liquid, liquid-gas mixture? What bubbles are you writing about?
The 3D configuration of electrodes is not shown. Do you use 36 electrodes?
What software (FEM) did you use for capacitance measurement simulation?
The paper lacks a report about SNR used in the capacitance measurement simulation.
Line 329 "our first artificial neural network was totally free from noise". So, in my opinion, the article presents an example of poor scientific research methodology.
Figure 5. The plot shows that stopping training at the 600th epoch is premature.
Figure 6. In radiology, an axial slice is a slice that divides the body into upper and lower halves. Axial slice means a slice perpendicular to the Z axis of the scanner. Do you really show the axial slices of the fluidized bed? What is the image matrix size?
Why don't you show cross-sections in other planes?
256. Experimental generated training database. In line 102, you criticize the use of the reconstruction algorithm to generate the training dataset as being biased by the algorithm (supervised learning). Your reinforcement learning approach is also biased by the forward problem operator. Can't you see this?
There is no description of the measurement system used for the measurements. The parameters of the measurement system are not provided.
Author Response

(The authors gave the same response as above.)

Reviewer 3 Report
Comments and Suggestions for Authors
Background:
The manuscript addresses applying Machine Learning (ML) to ECVT data toward more accurate and faster reconstruction. This topic is interesting as it focuses on an emerging trend of using ML to solve the complex ECVT reconstruction in an efficient manner. The authors used two methods:
First method: they used simulated fluidized bed data to generate realistic flow conditions. Then they used the sensitivity matrix to solve the forward problem and generate a capacitance vector for each case of solid flow distribution. The algorithm was then trained to match both together.
Second method: they used measured ECVT capacitance data and mapped it to an output image. The image was optimized by minimizing the error between the forward solution of the guessed image and the measured ECVT:
Review:
The main contribution to this work is the use of fluidized bed simulations to create realistic solids flow distributions. Adding fluid dynamics simulations to the ECVT reconstruction problem is a welcomed step in the right direction. However, the authors did not emphasize the error in the sensitivity matrix model that will propagate each time the forward or inverse solution is applied. The relation between capacitance measurements and solids distribution is complex and nonlinear. The sensitivity matrix itself is a source of error that is difficult to eliminate. Using the sensitivity matrix in both forward and inverse solutions imposes an inherit limitation to the accuracy of the final result. This is why ML, in this form, will not perform better than other reconstruction techniques, in terms of accuracy, as all are bounded by those inherit errors.
Recommendation:
I recommend accepting the manuscript after a major revision, the revision should address the following points:
1- Focus the paper on integrating the fluidized bed simulations, as this is the main contribution.
2- Explain the limitations of the forward solution using the sensitivity matrix model and explain that the accuracy of ML reconstruction is expected to be in par with other reconstruction techniques dues to the sensitivity matrix limitation. The ML approach here is mainly to speed up the reconstruction process, which the authors demonstrated successfully.
Author Response

(The authors gave the same response as above.)

Reviewer 4 Report
Comments and Suggestions for Authors
The article 'A methodology for generating a training database for machine learning based image reconstruction algorithms for ECVT: Application to fluidized bed reactors' presents two ways of generating training database for ANN, which were used for generating ECVT images during fluidization.
The article is written correctly, in accordance with the journal's requirements. The presented literature references are current, but their number could be greater.
My comments are mainly focused on the construction of the neural network, because its use in the described case is most advisable.
1. The first one concerns the number of hidden layers - in the text in lines 171-174 there are 3 hidden layers, while in the drawings (Fig. 4) there is one.
2. why was this particular network structure chosen (3 hidden layers)? Has this value been optimized? if so, how?
3. the paper presents RMSE, but there is no discussion on the size of the obtained errors - are they acceptable? what does it look like in the case of other methods used so far?
4. the presented approach has no practical value - the reader cannot reproduce or use this network. There is no data in the supplementary materials! Training databases or obtained weights in individual neurons of the constructed network should be placed there. Only then can you check whether the results obtained are correct. Other scientists can take advantage of them and use them for their own applications or develop them.
Taking into account especially comment No. 4, I am sending the article to further stages of evaluation after making major corrections.
Author Response

(The authors gave the same response as above.)

Round 2
Reviewer 1 Report
Comments and Suggestions for Authors
All queries have been addressed.
Comments on the Quality of English LanguageThe manuscript could use minor language editing, such as line 362, page 17: "to compared against" etc.
Author Response
Thank you for this second review.
Reviewer 2 Report
Comments and Suggestions for Authors
I can not recommend the paper for publication because the authors have not made the expected improvement in their work.
The authors responded to my remarks widely but have not made appropriate modifications to the text.
I agree with some answers, but I can not agree with the error claims, for example, with the description of linear approximation in ECT. The authors’ understanding of the problem is insufficient. The description of linear approximation is misleading. The following sentence is incorrect: "This matrix is known because it only depends on the system geometry, the sensors characteristics and the desired resolution."
Most of the ECT community does not use the name ECVT. The idea of confining yourself to small groups of specialists is far from open science. But I can accept the name you like.
The authors added a description of the data acquisition system, but it is insufficient. The electrodes of the sensor and their geometry are not described sufficiently. There is no drawing. The position and size of electrodes are very important in 3D ECT.
“To obtain the experimental data, we placed our ECVT system in a Plexiglas column of 10 cm internal diameter and 1m height.” How were electrodes placed in this column? It is not clear what the authors mean by “our ECVT system” here.
The authors do not show the simulated data. The example of permittivity distribution and the corresponding capacitance measurements should be shown. The samples of the training data set should be presented. There are known methods of presenting 3D data in tomography. In the ECT community, it is known how to show capacitance measurements in ECT.
I do not understand why it is possible to show good images in the answers to the reviewer, but is not possible to show them in the paper.
The author wrote about 36 capacitance sensors. Do they mean 36 electrodes or 36 electrode pairs?
The plots of capacitance measurements from the uniform and nonuniform permittivity (with the bubbles) should be shown. This will show us the sensitivity of the measurements, which is a problem in 3D ECT.
I emphasize my opinion that a paragraph explaining what a neural network is is unnecessary in a scientific article. The same concerns Figures 1 and 2. Instead, the authors should present their neural network configuration and better report their methods.
I agree that the capacitance measurements can be calculated using the linear approximation. How the sensitivity matrix is calculated?
The noise can be added to the capacitance data after simulation. I can accept the training on the data without the noise, but the validation should be performed on the data with the noise added. This is not a separate issue, as we can not separate the noise from the measurements.
Typically, a dataset is divided into training, testing, and validation parts, or, in machine learning terminology, training, validation, and testing parts. Why do you only use two of these subsets? There are no appropriate comments on this issue.
The size of the training data set is not discussed. From my experience, it seems that the size of the training data set is too small compared to the size of the problem (the number of output elements) to give good results.
I agree with the response of the authors to the comment about the premature stopping of the learning algorithm, but the paper lacks this discussion. The reader does not know why the algorithm is stopped in that step.
The authors should remove the lecture on neural networks and report their results better instead. This applies to several places in this article.
Reviewer 3 Report
Comments and Suggestions for Authors
The authors implemented changes to address comments. The paper can be published.
Author Response
Thank you for this second review.
Round 3
Reviewer 2 Report
Comments and Suggestions for Authors
I am still not convinced about the scientific level of the article. Therefore, I cannot recommend the article for publication.
The authors have made some corrections and improvements to the text, making it more transparent. They also finally removed the incorrect theorem about the sensitivity matrix.
Unfortunately, the article contains basic, book-like content that is inappropriate for a scientific article. How badly educated do the authors think the chemical engineering community is? Meanwhile, their own experiments are not described well enough. There are no photos of the measurement setup or experiments. The simulated or measured data are not shown. It doesn't matter whether they are normalized or not.
The quality of the obtained images is questionable. The authors only show images for single cases, arguing that the average RMS error is much more important. Unfortunately, even the selected case demonstrates that the obtained reconstruction quality is poor (Fig. 18).
The instantaneous value of the mean squared error is also alarmingly large. Anything with an error greater than LBP means a very large image reconstruction error. My experience shows that with a well-selected training set, ANN outperforms LBP for most images and does not behave like a generator. Perhaps the poor quality of the obtained images results from the quality of the measured capacitances (simulated or measured). Unfortunately, the authors do not want to show these data. However, the problem may lie also in the data generation or training stage. No example elements of the training set are shown.
The conclusions drawn are unjustified. For example, "The results obtained showed an excellent agreement" is not covered by the data. Does this sentence apply to one selected image or the MSE?
The following phrase used in the summary is not clear: "the most complex patterns present in a fluidized bed." The authors do not show examples of complex patterns in the article, as they certainly claim that all this is well-known in the chemical engineering community.
The remaining conclusions are mostly obvious, for example: “however the ANN can process data much faster.” This is an assumption, not a conclusion. We know it from the books. Because of this, we are playing with ANNs.
Some conclusions are false. "Our study showed that the artificial neural network was as accurate as the best algorithms found in the literature”. There are algorithms that perform much better than MOIRT or ILBP discussed here. This is another example of false or negligent claims continuing to appear in the article.
Author Response
Further to our email exchanges with Quinta Ding (01/31/2024). We are resubmitting the latest version of the article without responding to reviewer 2's final remarks. Final decision to be made by the editor.